# Regulation of Carcinogenesis by Sensory Neurons and Neuromediators

**DOI:** 10.3390/cancers14092333

**Published:** 2022-05-09

**Authors:** Nuray Erin, Galina V. Shurin, James H. Baraldi, Michael R. Shurin

**Affiliations:** 1Department of Medical Pharmacology, Immunopharmacology, and Immuno-Oncology Unit, School of Medicine, Akdeniz University, 07070 Antalya, Turkey; 2Department of Pathology, University of Pittsburgh Medical Center and University of Pittsburgh Cancer Institute, Pittsburgh, 15213 PA, USA; shuringv@upmc.edu (G.V.S.); shurinmr@upmc.edu (M.R.S.); 3Department of Neuroscience, University of Pittsburgh Medical Center and University of Pittsburgh Cancer Institute, Pittsburgh, 15213 PA, USA; james.baraldi@pitt.edu; 4Department of Immunology, University of Pittsburgh Medical Center and University of Pittsburgh Cancer Institute, Pittsburgh, 15213 PA, USA

**Keywords:** tumor innervation, neuro-immunology, neuropeptides, metastasis

## Abstract

**Simple Summary:**

Sensory nerve fibers extensively innervate the entire body. They are the first to sense danger signals, including the ones coming from newly formed cancer cells. Various studies have demonstrated that the inactivation of sensory nerve fibers as well as the vagus nerve enhances tumor growth and spread in models including breast, pancreatic, and gastric cancer. On the other hand, there are also contradictory findings that show the opposite, namely that the inactivation of nerve fibers inhibits tumor growth. These discrepancies are likely caused by the stage and the level of aggressiveness of the tumor model used. Hence, further studies are required to determine the factors involved in neuro-immunological mechanisms of tumor growth and spread.

**Abstract:**

Interactions between the immune system and the nervous system are crucial in maintaining homeostasis, and disturbances of these neuro-immune interactions may participate in carcinogenesis and metastasis. Nerve endings have been identified within solid tumors in humans and experimental animals. Although the involvement of the efferent sympathetic and parasympathetic innervation in carcinogenesis has been extensively investigated, the role of the afferent sensory neurons and the neuropeptides in tumor development, growth, and progression is recently appreciated. Similarly, current findings point to the significant role of Schwann cells as part of neuro-immune interactions. Hence, in this review, we mainly focus on local and systemic effects of sensory nerve activity as well as Schwann cells in carcinogenesis and metastasis. Specific denervation of vagal sensory nerve fibers, or vagotomy, in animal models, has been reported to markedly increase lung metastases of breast carcinoma as well as pancreatic and gastric tumor growth, with the formation of liver metastases demonstrating the protective role of vagal sensory fibers against cancer. Clinical studies have revealed that patients with gastric ulcers who have undergone a vagotomy have a greater risk of stomach, colorectal, biliary tract, and lung cancers. Protective effects of vagal activity have also been documented by epidemiological studies demonstrating that high vagal activity predicts longer survival rates in patients with colon, non-small cell lung, prostate, and breast cancers. However, several studies have reported that inhibition of sensory neuronal activity reduces the development of solid tumors, including prostate, gastric, pancreatic, head and neck, cervical, ovarian, and skin cancers. These contradictory findings are likely to be due to the post-nerve injury-induced activation of systemic sensory fibers, the level of aggressiveness of the tumor model used, and the local heterogeneity of sensory fibers. As the aggressiveness of the tumor model and the level of the inflammatory response increase, the protective role of sensory nerve fibers is apparent and might be mostly due to systemic alterations in the neuro-immune response. Hence, more insights into inductive and permissive mechanisms, such as systemic, cellular neuro-immunological mechanisms of carcinogenesis and metastasis formation, are needed to understand the role of sensory neurons in tumor growth and spread.

## 1. Introduction

The relationship between the immune system and the nervous system in driving carcinogenesis is well established [1]. Materialization of a malignant neoplasm coincides with inflammation, an immune response that the nervous system may regulate via neuromediators and neuroglia. Which appears first, the tumor or the inflammation, may vary: oncogenic activity may precede inflammation, or inflammation may promote conditions conducive to tumorigenesis [2]. Regardless, the cellular and molecular immune and nervous constituents of the tumor microenvironment synergize an overexpression of mediators of inflammation, such as inflammatory cytokines, with the recruitment of the innate immune response.

Natural killer cells, for instance, release a diversity of pro-inflammatory cytokines. The neuropeptide substance P (SP) has been shown both to potentiate [3] and to attenuate [4] these cells’ cytotoxicity, in both instances by acting on the neurokinin-1 receptor. This receptor appears not only on natural killer cells specifically and immune cells broadly but also on neurons [5] and cancer cells [6]. This distribution of the neurokinin-1 receptor across the three eponymous systems of neuro-immuno-oncology suggests an inherent integration in both the etiology of and the response to tumor development.

Interactions between the immune system and the nervous system are crucial in maintaining homeostasis such that abnormal activation or disturbances of these neuro-immune interactions have pathological consequences [7,8]. The best examples of neuro-immune interactions are provided by studies examining the effects of chronic stress. Disproportionate physical and emotional stresses induce pathological activation/inhibition of neuro-immune pathways, leading to chronic inflammatory diseases [9]. Chronic stress-induced alterations in immune responses have detrimental consequences for cancer patients due to suppression of protective immunity, induction or exacerbation of chronic inflammation, and enhancement of immuno-suppressive mechanisms [10,11]. Psychosocial interventions to cope with stress, if delivered early in the disease, have been associated with a significant reduction in cancer mortality [12].

Neuro-immune interactions occur in both the central nervous system (CNS) and peripheral nervous system (PNS) [7,8]. The nervous system outside the brain and spinal cord is called the peripheral nervous system, which is divided into the autonomic and somatic branches. The autonomic nervous system controls internal homeostasis via the sympathetic and the parasympathetic divisions. The somatic nervous system includes the sensory and somatosensory parts and anatomically consists of sensory, somatic, and mixed nerves. Most tissues in the body are innervated to varying degrees by both sensory and autonomic nerves. In accordance with neuronal architecture, multiple components of neuro-immune interactions exist and include the adrenergic system (catecholamines by autonomic nerves and the adrenal glands), hypothalamic–pituitary–adrenal (HPA) axis (cortisone), and sensory nerves (neuropeptides) [13,14,15,16,17]. Among peripheral nerves, the vagus nerve, which mostly carries sensory nerve fibers (over 80%), plays a special role in neuro-immune regulation through the cholinergic anti-inflammatory pathway [18,19,20,21,22]. The nervous system not only alters the immune response by providing neuromediators released by nerve fibers but also by carrying glial cells along the axons. Peripheral glial cells are called Schwann cells, and recent findings point to the important immunomodulatory role of these cells under various conditions, including cancer [23,24].

The microenvironment of cancer cells includes immune cells, newly formed blood vessels, fibroblasts, and components of the nervous system. Intra-tumoral nerve fibers are present in various carcinomas including breast, prostate, colorectal, and pancreatic cancers [25]. The origins or types of the intra-tumoral nerve fibers are diverse and include sympathetic, parasympathetic, and sensory nerves, and, depending on the cancer type, the composition and origin of nerve fibers may change [25,26]. The tumor microenvironment also includes immunologically active neuropeptides/neuromediators such as calcitonin gene-related peptide (CGRP), substance P, neurotensin, and nerve growth factor (NGF) [27,28,29,30,31]. Intra-tumoral fibers and neuropeptides have diverse functions that may enhance or decrease tumor progression [25,28,32,33,34]. For example, the presence of sympathetic nerves in the tumor stroma of breast cancer patients may promote disease progression, and parasympathetic nerves are likely to do the opposite [35]. Although tumor-promoting effects of the sympathetic activity are well appreciated [25,32,33,34], conflicting results regarding the direct and indirect effects of sensory neurons and the vagus nerve on carcinogenesis and metastasis are present. Hence, in this review, we mainly focus on the sensory component of the nervous system, which is present with its Schwann cells, in the development and progression of cancer.

## 2. Sensory Neurons

Functionally, the PNS is divided into motor (efferent) and sensory (afferent) nerves. Primary afferent (sensory) neurons have their cell bodies located in cranial and spinal ganglia. These nerve fibers carry information from target organs to the central nervous system [36,37,38]. Sensory neurons are categorized into three groups based on their conduction velocity and diameter. Thin unmyelinated fibers are called C-fibers, and they contain polymodal nociceptors and a smaller portion of mechanoreceptors. The other two groups include small A-fibers and large A-fibers, and both are myelinated. Small A-fibers transmit both nociceptive and non-nociceptive information, while the large A-fibers transmit non-nociceptive information from the muscles and joints. Sensory neurons that are categorized as spinal have their cell bodies in the dorsal root ganglia. Cell bodies of visceral sensory nerve fibers, which are mainly carried by the vagus nerve, are located in the nodose and jugular ganglia and project to the brainstem [39,40].

The nerve endings of somatosensory neurons may be the receptors themselves or may be connected with special sensory structures, like Pacinian, Ruffini, Meissner’s corpuscles, or Krause’s organs [36,40]. Sensory endings that are not specialized morphologically are referred to as “free endings”. Sensory nerve endings are polymodal, activated by different types of stimuli, and classically involved in the sensation of pain, as demonstrated in somatic cutaneous afferents [41]. Visceral afferents, however, are different since their principal role is to maintain homeostasis of the internal environment by controlling circulation, temperature, digestion, respiration, and the immune response. Pain sensation associated with visceral afferents is often poorly defined and projects to somatic structures such as skin and muscle [36].

There are different types of sensory neurons that vary in their location, their structure, and the stimuli to which they respond. Visceral sensory neurons innervate essentially all internal organs, including lymphoid tissue, epithelia, glands, dental pulp, blood vessels, mast cells, secretory cells, and smooth muscle. Sensory neurons also innervate the joints, muscles, and skin [38,42,43,44,45]. These nerve endings are activated and sensitized by various insults, including pathogens, tissue damage, protons produced in acidic tissue upon inflammation, and various inflammatory mediators [46]. Hence, complex anatomical organization, diverse target innervation, multiple chemical messengers, and a wide range of targets for the sensory neuromediators prove their broad functional roles. Effector roles of peripheral endings of afferent neurons are not confined to modulation of local blood flow; they also extend to the modulation of the immune response to a wide range of stimuli, including cancerous cells [46,47]. The plasticity of spinal neural circuits also supports these effector roles, as certain cues, like peripheral inflammation, induce neuropeptide expression in the DRG as well as hyperalgesia [36]. Nerve injury also represents a good example of the plasticity of sensory nerve endings whereby the phenotypic expressions of primary afferents, and consequently effector functions mediated by neuropeptides, are altered [42,48,49,50,51].

## 3. Capsaicin-Sensitive Sensory Neurons and Their Responses to Pathogens and Cancer

Capsaicin-sensitive sensory neurons are sensitive to capsaicin, the pungent ingredient in hot chili peppers, that selectively produces functional blockade [52,53]. Capsaicin stimulates the transient receptor potential vanilloid 1 (TRPV1) and ankyrin 1 (TRPA1) receptors [54] that are expressed mainly by unmyelinated fibers, and these receptors are also found in some small-diameter myelinated sensory neurons as described above [55]. Capsaicin at low doses activates unmyelinated axons, whereas at high doses it induces the inactivation of unmyelinated fibers. Activation of the TRPV1 receptor leads to an influx of calcium and sodium cations, and excessive activation triggers an excitotoxic effect [38,47,48,56,57,58,59,60]. TRPV1 receptors are not found in large-diameter sensory neurons, motor neurons, or sympathetic neurons [61,62]. Hence, only unmyelinated and small-diameter myelinated sensory neurons are destroyed after high-dose capsaicin injection, and large sensory afferents, as well as sympathetic fibers and their motor functions, are not affected [63,64,65].

Capsaicin-sensitive sensory nerves do not only have sensory (afferent) functions but also local effector functions [36,66] through releasing neuroactive peptides. Neuropeptides such as calcitonin gene-related peptide and substance P are synthesized in unmyelinated sensory neurons and released from their peripheral terminals upon activation by local factors. Accordingly, capsaicin treatment depletes CGRP and SP in peripheral nerve endings, while it does not alter the neuropeptide levels in the CNS [65,67,68,69].

The best example of locally mediated effector function is neurogenic inflammation, which is mediated by the release of sensory neuropeptides locally upon activation. These peptides include SP, CGRP, pituitary adenylate cyclase-activating polypeptide (PACAP), and vasoactive intestinal polypeptide (VIP). Neurogenic inflammation is characterized by vasodilation and immune cell recruitment [36,70]. Somatic and visceral afferents induce local tissue responses that spread to both central and peripheral terminals (axon reflex) and also establish plasticity in response to inflammation and injury [48,71]. The axon reflex is also called the flare response, which is accomplished by a bifurcated axon. Activation at one limb of the axon leads to activation of its collateral axons and causes the release of neuropeptides such as CGRP and SP, altering the innervated cells [39,72,73]. These neuropeptides have receptors on cells that are not directly in contact with nerve fibers, further validating the effector functions of afferent fibers [38,48]. Flare due to vasodilatation (reddening), which spreads beyond the pin-point injury of the skin, is the result of axon reflexes amongst the arborizing collaterals of nerve fibers. Nerve impulses travel centrally to the branching points and then move antidromically to the other branches coming back to the skin. Here, periarteriolar branches of sensory neurons can release vasoactive neuropeptides, (e.g., SP and CGRP) and thereby cause arteriolar dilatation.

However, the exact roles of capsaicin-sensitive sensory nerves and capsaicin receptors in cellular neuro-immunological mechanisms of carcinogenesis and metastasis formation in the tumor milieu are not well known. A comprehensive understanding of these pathways has clear consequences for mechanism-based drug development for controlling the capsaicin receptors in cancer.

Both the nervous and immune systems play important roles in regulating several mechanisms essential for protection against external threats and acute stressors and in maintaining physiological homeostasis. Sensory neurons are activated by a wide range of stimuli, including pathogens, danger signals, and inflammatory molecules. The response kinetics of the neurons to pathogens are much faster than those of the immune cells, as the former occurs within milliseconds compared to hours for the latter. Hence, the early response of the nervous system is involved in the coordination of host defenses in eliminating threats and facilitating tissue repair [74,75,76]. For example, the pathogen *Staphylococcus aureus* activates axon reflexes, causing the release of CGRP, somatostatin, and galanin in an antidromic manner. These neuropeptides activate their receptors found on mast cells, monocytes, neutrophils, and macrophages and inhibit *S. aureus*-induced excessive activation of the innate immune system, leading to tonic anti-inflammatory effects [77,78] (Figure 1).

Capsaicin-sensitive neurons have a protective effect against acute injury in the gastric mucosa, as well as in the colon, such that inactivation of these nerve fibers results in increased inflammation and tissue damage in the immune-complex model of colitis in rabbits [79]. Similarly, inactivation of the vagus nerve, which carries the preponderance of visceral capsaicin-sensitive C-fibers, worsens dextran sulfate sodium (DSS)-induced colitis in mice, a consequence of the decreased potential of antigen-presenting cells to induce regulatory T cells [40,80,81]. Sensory nerves in immunized animals also decrease antigen transit to the lymph nodes [82]. Direct stimulation of the local neural plexus of the lymph nodes reduces antigen trafficking to the lymph nodes, a phenomenon that is likely due to antigen-specific activation of nociceptors via Fc receptors (FcRs). Functional FcRs are found in a subpopulation of sensory neurons, and stimulation of FcRs by antigen-antibody complexes activates nerve fibers [83]. Hence, activation of sensory neurons limits the excessive immune activation and inflammatory response protecting the organism. These results also show that the lymphatic spread of a pathogen, or a metastatic cancer cell is also controlled by sensory neurons innervating the lymph nodes [82].

Given the important role of sensory nerve activity in homeostasis and protection against internal and external insults, it is conceivable that damage to afferent C-fibers may have serious adverse consequences. A good example is an infection with *Mycobacterium leprae*, the causative bacterium of leprosy. When the bacilli invade axons and Schwann cells, the principal glial cells of the PNS, they induce serious nerve damage [84]. Therefore, the progressive depletion of SP and other neuropeptides from the skin is observed in leprosy [85]. The destruction of sensory C-fibers and sympathetic innervation annihilates anti-inflammatory feedback circuits. Diminishing levels of neuropeptides and changes in the cytokine profile modulate the inflammatory site by affecting the sensitivity of infiltrating T cells and inducing their resistance to anti-inflammatory steroids [86]. Hence, the destruction of these fibers in leprosy diminishes the activity of neural feedback pathways, including hypothalamic–pituitary–adrenal (HPA) axis activation, leading to the augmented inflammatory activity and increased severity of the disease [86,87]. Sensory nerve-induced HPA activation is likely due to SP-induced release of interleukin-1 (IL-1) from various cells, such as keratinocytes [88] and macrophages [89]. Leprosy-induced depletion of nerve growth factor, which occurs before the damage of peptidergic nerves, seems to be responsible for the loss of nerve fibers: NGF participates in the maintenance of the peptidergic nerves, raising their SP content [90].

Similar nerve damage can also occur due to malignant tumor growth. Cancerous cells and neurons intermingle, and in some cases, there is a complete loss of the neural elements at the tumor site [91]. A significant nerve injury response has been reported during the growth of human melanoma [92]. Another pathway of neuron–tumor cell interaction is observed during perineural invasion (PNI) by cancer of the space surrounding a nerve. PNI is a marker of poor prognosis: it is associated with increased recurrence rates and decreased patient survival rates [93,94,95,96]. PNI may be found in many cases of pancreatic cancers, head and neck cancers, prostate cancers, skin cancer, and colorectal cancer [96,97,98]. A phenomenon termed “neural remodeling”, characterized by the alterations in the morphology of the nerve has been postulated in PNI [99,100,101]. Perineural invasion results in nerve damage, including myelin damage and axon degeneration and loss [98,102]. In the perineural space, PNI induces reactive alterations in the morphology and function of the nerves. Morphological changes include changes in the thickness of the nerve trunk [100]. Electrophysiological evidence of nerve injury consists of reduced conduction velocity and an enhanced percentage of both mechanically insensitive and electrically unexcitable neurons [102]. Studies have revealed that some signaling molecules and pathways that are involved in PNI are also involved in pain generation [97]. PNI causes pain, paresthesia, and numbness, demonstrating the involvement of sensory nerve fibers. Recent data suggest that PNI-induced pain may be driven by nerve injury and peripheral sensitization in mechanoreceptors [102].

Destruction of sensory fibers is likely to proceed by the growth of new axons into the tumor tissue in a way similar to angiogenesis and termed neoneurogenesis [103]. The function and content of these newly formed nerve fibers are not presently known; given the symptoms, they are likely to be more pathological than physiological and could be considered abnormal neurogenesis. Patients with cancers with increased sensory nerve innervation complain of cancer pain and have poor outcomes [104].

## 4. Sensory Nerve Activity and the Antitumor Immune Response

Neurogenic inflammation, as described above, is a well-known example of the locally mediated effector function of capsaicin-sensitive sensory nerves, a function mainly mediated by SP and CGRP and characterized by an acute inflammatory response [36,59,105]. An acute inflammatory response is required for the elimination of pathogens, including cancerous cells [106,107]. The role of C-afferents in inflammation, however, is not limited to neurogenic inflammation. Many studies document the anti-inflammatory consequences of sensory nerve activation, which seems to be a defense system for resolving inflammation and promoting tissue repair [49,108]. Cancer is considered to consist of “wounds that do not heal” [109,110], and chronic inflammation is involved in the initiation, growth, progression, and metastasis of cancer [111,112,113,114,115,116]. Consequently, activation of sensory nerves and release of anti-inflammatory and immune-regulatory neuropeptides may induce the final phase of the healing process in cancer, a phase that might be associated with tumor regression. For example, systemic inactivation of sensory neurons using a neurotoxic dose of capsaicin increases lung and heart metastasis in a syngeneic murine breast carcinoma model [47,56,57]. Neurotoxic effects of capsaicin are reversible when used in adult mice [63], and when sensory nerves are allowed to regenerate for three weeks metastatic potential returns to baseline levels, demonstrating the protective effects of capsaicin-sensitive afferents.

A significant number of visceral afferent C-fibers are found in the vagus nerve and convey sensory information from most visceral organs to the brain [117]. The vagus nerve innervates internal organs such as the airways, lungs, gastrointestinal tract, pancreas, liver, bile ducts, and portal vein [39,40,118]. It was shown that unilateral left cervical vagotomy or perivagal capsaicin treatment, either of which inactivates afferent unmyelinated C-fibers, markedly increased lung metastases of breast carcinoma, demonstrating a protective role of vagal sensory C-fibers [56,119,120]. Similar results were obtained by using a murine orthotopic pancreatic cancer model in which subdiaphragmatic vagotomy results in increased tumor growth and impaired survival accompanied by increased inflammation [121]. Bilateral subdiaphragmatic vagotomy also enhanced experimental carcinogen-induced pancreatic cancer [122] and gastric cancer [123,124]. Accordingly, clinical studies demonstrate that patients with gastric ulcers who have undergone a vagotomy have a greater risk of stomach, colorectal, biliary tract, and lung cancers [125,126]. Inactivation of the hepatic vagus aggravates the development of liver metastases [127]. Protective effects of vagal activity against cancer have also been documented by epidemiological studies, demonstrating that high vagal activity, indexed by heart-rate variability [128], predicts longer survival rates in patients with colon, non-small cell lung, prostate, and breast cancers [129,130]. The protective effects of vagal activity are linked to the inhibition of inflammation in pancreatic cancer [131]. The anti-inflammatory effects of the vagus nerve are partly due to activation of the efferent arm by vagal sensory afferents; this is called the cholinergic anti-inflammatory pathway [132,133,134]. Activation of the nicotinic acetylcholine receptors (nAChRs) on immune cells such as macrophages [133,135,136,137] and dendritic cells is likely to mediate the anti-inflammatory effect of the vagus nerve [138,139]. The anti-inflammatory effects of vagus nerve activity may underlie the protective effects of systemic vagus nerve activity during cancer progression [132,140,141].

Recent findings have demonstrated that peptidergic sensory fibers also contain molecules that regulate nicotinic cholinergic transmission, such as CGRP and secreted mammalian Ly6/urokinase plasminogen activator receptor-related protein-1 (SLURP-1) [142]. SLURP-1 is an endogenous ligand of the α7 subunit of nicotinic acetylcholine receptors, and these receptors are considered the mediators of the anti-inflammatory response observed following vagus nerve activation [140,143]. Moriwaki et al. demonstrated that SLURP-1 could be found in primary peptidergic sensory neurons and acts as a cholinomimetic on nAChRs [142]. Nicotinic cholinergic transmission can also be potentiated by CGRP. By activating PKA, PI3 kinase, and PKC, CGRP enhances the effects of AChRs [144,145,146] and increases the expression of the α-subunit of the receptor [147,148]. Activation of CGRP receptors decreases the level of acetylcholinesterase, which inactivates acetylcholine, thus further enhancing cholinergic activity [147]. Peptidergic-cholinergic interactions seem to be bidirectional since the expression of the nAChR subunit α3 (CHRNA3) on the nociceptive afferents was demonstrated in neurons of which ~50% are peptidergic [149]. All of these findings demonstrate that sensory neuropeptides are involved in the cholinergic anti-inflammatory pathway.

## 5. Sensory Neuromediators

Of the sensory neuropeptides, SP and CGRP seem to be implicated as among the main mediators of immunomodulatory and antitumoral features of sensory neurons. CGRP is found in almost half of the somatic axons, and the majority of CGRP is localized on the C-fibers sensitive to capsaicin. Half of the CGRP-containing C-fibers also contain SP [150]. Therefore, SP and CGRP should play important roles in the regulatory effects of C-fibers on inflammation, the immune response, and cancer (Figure 2), and below we explain their roles in cancer and inflammation.

### 5.1. Substance P

Substance P is an undecapeptide of the tachykinin neuropeptide family and is broadly distributed in the CNS and PNS [151]. SP mediates its functions mainly by interacting with its chief receptor, the neurokinin-1 receptor (NK-1R) [152]. SP- and NK-1R-expressing neurons participate in pain, stress, and anxiety reactions. In neurons, SP is expressed in the soma and released through the exocytosis of the vesicles either at the axonal terminals or at the neuronal soma [153]. In addition, SP is secreted by immune cells such as macrophages, eosinophils, lymphocytes, and dendritic cells, and it is involved in neurogenic inflammation [154]. NK-1R is distributed over cytoplasmic membranes of many cell types, including neurons, glia, endothelial cells, fibroblasts, stem cells, and leukocytes. Thus, the SP/NK-1R system can influence or provoke many cellular processes. For instance, it mediates crosstalk between neurons and immune cells, and it modifies immune responses, including leukocyte activation, proliferation, and cytokine expression [155]. Mutually, cytokines can induce the expression of SP and NK-1R on different cells [156,157].

Nerves containing SP were reported to innervate primary (thymus and bone-marrow) and secondary (spleen, lymph nodes, tonsils, and the gut-associated lymphoid tissue) lymphoid organs [158,159,160]. In line with this innervation pattern, NK-1R was found in immune cells, including T lymphocytes [161], B lymphocytes [162], macrophages [163], dendritic cells [164], neutrophils [165], mast cells [166], and natural killer cells [3]. In addition to neurogenic inflammation, SP stimulates an immune response and, at an acute phase, induces the release of inflammatory cytokines, increasing the cytotoxicity of immune effectors. Specifically, SP regulates the influx of neutrophils and enhances their phagocytic activity in inflamed tissues [160,167] in an NK-1R-dependent manner. In vivo stimulation of NK-1R potentiates the immunostimulatory functions of skin-resident Langerhans cells to induce antigen-specific type 1 immunity in a mouse model [168]. Activation of NK-1R on dendritic cells upregulates the production of IL-12 and induces type 1 immunity [169]. SP also has a role in maintaining tissue-resident dendritic cell populations under homeostatic conditions [71,160,170].

SP dose-dependently increases the migration of NK cells [3] and induces interferon-gamma (IFN-γ) [171] and IL-12 secretion [172] by immune cells. For instance, SP enhances lymphokine-activated killer cell cytotoxicity as well as NK cell cytotoxicity, augmenting IL-12 production by macrophages [173]. Similarly, SP acting through NK-1R on dendritic cells promotes type 1 immunity, IL-12 secretion, and dendritic cell maturation, which collectively enhance the efficiency of dendritic cell vaccines [56,169]. Interestingly, IL-12 induces SP expression in splenic macrophages, suggesting that IL-12 and SP regulate each other’s expression in murine macrophages [174]. In addition to IL-12, IL-23 also regulates SP levels in mouse macrophages, which can be inhibited by transforming growth factor beta (TGF-β) [160,175]. Moreover, IL-1, IL-4, and IFN-γ induce NK-1R expression in macrophages [152,176,177]. Substance P also enhances phagocytosis in murine peritoneal macrophages via its N-terminus [167], induces oxidative burst, and stimulates synthesis and release of arachidonic acid metabolites and toxic oxygen radicals [152,178,179].

Substance P stimulates human T cell proliferation in vitro through upregulation of IL-2 expression [161,180,181,182,183,184,185] and IL-12-, IL-18-, and tumor necrosis factor alpha (TNF-α)-induced NK-1R expression on T cells [186,187]. IL-10 and TGF-β, on the other hand, prevent NK-1R expression [187,188]. SP enhances immunoglobulin secretion in murine Peyer’s patches, splenic lymphocytes, and mesenteric lymph nodes in an isotype-specific manner (particularly IgA) [189]. Administration of SP during the primary immune response amplifies the secondary immune response by activating CD8+ T lymphocytes [185]. Under certain conditions, SP abolishes the immunosuppressive activity of regulatory T (Treg) cells [190], demonstrating the in vivo ability of SP to regulate Treg cell function. SP may also promote the generation of memory CD8+ T cells during the primary immune reaction by enhancing antigen-presenting cell function [185].

These findings demonstrate the ability of SP to activate cytotoxic immunity against pathogens and, possibly, transformed cells such as pre-cancerous cells. Thus, continuous treatment with relatively low doses of SP markedly enhances the antitumor effects of ionizing radiation in a breast carcinoma model, inducing a complete response in 50% of mice [56,191]. This antitumor effect of SP is likely to be mediated by an increased antitumor immune response since SP treatment decreases the number of tumor-infiltrating myeloid-derived suppressor cells (MDSCs) as well as the release of TNF-α while enhancing IFN-γ secretion from leukocytes [191].

Unbound SP can be degraded by the cell-surface metallo-endopeptidase neprilysin and ADAM-10 (a disintegrin and metalloproteinase domain-containing protein 10) into peptides, which may have differential functions [12,117,118,119]. In the tumor tissue, cleavage of SP by ADAM-10 results in the generation of antitumoral bioactive fragments [12,119].

As stated above, cancer is considered to consist of “wounds that do not heal” [109,110]. The healing properties of SP are best documented in the gastrointestinal tract, cornea, and skin wounds. Mice that lack NK-1R had markedly worsened colitis in several colitis models, especially during the chronic phase, demonstrating that SP may enhance mucosal healing during an inflammatory response [192]. Accordingly, SP via activation of NK-1R promotes a diabetic corneal epithelial wound [193]. Prolonged inflammation causes irreversible damage by hindering regeneration. SP treatment decreases injury-induced inflammatory responses by increasing IL-10 and decreasing TNF-α levels after wounding [194]. SP may promote tissue repair by mobilizing bone marrow mesenchymal stem cells to become involved in the repair process [195,196], and application of SP multiple times accelerates wound closure in both diabetic and non-diabetic hosts [197,198]. Other studies have also documented anti-inflammatory effects of SP during wound healing: SP increases alternatively activated type M2 macrophage and TREG cell numbers and IL-10 levels while it decreases TNF-α levels [199,200,201,202]. These findings are in contrast with documented pro-inflammatory effects of SP, demonstrating that the influences of SP depend on the timing and state of the pathological process, e.g., chronic inflammatory conditions or during carcinogenesis [56]. Thus, it is likely that SP may inhibit excessive inflammation that enhances cytotoxic immunity in advanced carcinoma [56].

### 5.2. CGRP in Inflammation and Immune Regulation

CGRP is a member of the calcitonin family of peptides, which also includes the calcitonin receptor-stimulating peptides, calcitonin, adrenomedullin, amylin, and intermedin [203]. Two forms of CGRP exist in humans: CGRP alpha (α-CGRP, CGRP1) and CGRP beta (β-CGRP, CGRP2). The receptors of the CGRP family are called calcitonin receptors (CTRs) and calcitonin receptor-like receptors (CRLRs) [204]. CGRP is found in both peripheral and central neurons and is involved in the vascular supply of peripheral nerves, nociception, cardiovascular homeostasis, ingestion, and modulation of the autonomic nervous system [205,206].

The anti-inflammatory effects of CGRP have been well documented. CGRP markedly decreases lipopolysaccharide (LPS)-induced upregulation of blood neutrophils, TNF-α secretion, and tissue accumulation of neutrophils, and it enhances IL-10 production by peritoneal macrophages while inhibiting TNF-α secretion [108,207,208,209]. CGRP treatment inhibits LPS-induced production of chemokines, such as neutrophil chemotactic factors CXCL1 and CXCL8, and the monocyte and dendritic cell chemoattractant CCL2 by endothelial cells [71,210]. These effects are likely to be through the inhibition of nuclear factor kappa B (NF-κB) activation [210,211] or by inhibition of the phosphorylated inhibitor of NF-κB kinase β (p-Iκκβ) [212]. Cellular effects of CGRP may also be mediated by stimulation of adenylate cyclase and accumulation of cyclic adenosine monophosphate (cAMP) through the CGRP receptor [213].

CGRP is a potent arterial and venous vasodilator that effectively dilates microvessels [214]; this dilatation may play a role in the anti-inflammatory/protective effects of CGRP. Schneider et al. reported that pancreatic morphologic damage due to severe necrotizing pancreatitis is inhibited by CGRP treatment in vivo, and this protection is due to the inhibition of microcirculatory disturbances as well as inflammation [215]. Hepatic inflammatory responses due to ischemia-reperfusion injury are markedly higher in congenitally CGRP-deficient mice [108,216]. Administration of CGRP was also shown to prevent inflammation following an inflammatory stimulus in the skin in vivo, an occurrence due to inhibition of chemokine release by stimulated endothelial cells. This is likely to be physiologically important since cutaneous blood vessels are innervated by sensory nerves [208]. Interestingly, ovariectomy decreases the expression of CGRP, making the mucosa more prone to stress-induced injury, and, accordingly, the expression of CGRP is enhanced by estrogen in DRG cells [217,218]. It seems that there is a feedback mechanism in which inflammatory cytokines limit their secretion by increasing CGRP release. TNF-α activates C-fibers at a 5–100 pg/mL concentration, inducing the release of neuropeptides, including CGRP, but at higher concentrations, the effect is diminished [219]. These effects of cytokines are likely to be mediated by prostaglandins [220]. Prostaglandins facilitate CGRP release from rat DRG cells through activation of the cAMP transduction cascade [221,222]. Hence, inactivation or disturbance of sensory neurons by invading cancer cells can inhibit this negative feedback loop of inflammatory cytokines, leading instead to chronic inflammation, cancer progression, and metastases.

## 6. Glial Cells, Cancer, and Sensory Neuromediators

In the PNS, all axons are surrounded by specialized glial cells known as Schwann cells or neurolemmocytes, which are critically engaged in the development, operation, protection, and regeneration of peripheral nerves. They can be classified into two general types: myelinating and nonmyelinating cells. In general, small-diameter axons are unmyelinated and surrounded by Schwann cells. This type of nerve fiber is found in the parasympathetic and sympathetic nervous systems and in the majority of peripheral sensory fibers. For instance, nociceptive, thermoceptive, and pruriceptive, (i.e., itch perceptive) neurons are either unmyelinated (C-fiber) or lightly myelinated (Aδ) sensory neurons [223]. On the other hand, large-diameter neurons, group A and B fibers, are wrapped by multiple concentric layers of the Schwann cell plasma membrane—a myelin sheath, forming myelinated nerve fibers. They are found mostly in motor and sensory neurons.

Schwann cells, as the major glial cell type in the PNS, play an active role in peripheral nerve regeneration after axon damage. Damage to the neuronal body first initiates Wal-lerian degeneration—the catabolic process of axon degeneration distal to the site of the in-jury and orchestrated by Schwann cells. Wallerian degeneration involves the activation of Schwann cells associated with their denervation, dedifferentiation, and proliferation. They also attract macrophages to remove axon fragments, clear debris, and assist in myelin breakdown. [224]. Genetic reprogramming of Schwann cells leading to their dedifferentiation further transforms them into a repair phenotype to contribute to the concomitant nerve regeneration. Schwann cells have two crucial functions for peripheral nerve repair: creating a microenvironment to sustain the nerve regeneration and producing the guidance cues for the regenerating axon sprout as it grows. Secretome released by Schwann cells and macrophages and containing cytokines and growth factors, such as VEGF, TGF-β, IL-1β, IL-6, IL-8, and IL-15 [225], ensures that angiogenesis, remyelination, and axon regeneration occur sequentially in the process of nerve regeneration.

Schwann cells that are of the non-myelinating phenotype and associated with uninjured C-fibers are also activated upon injury, releasing diffusible factors, which activate nerve fibers [109,212]. Application of pro-inflammatory cytokines, such as TNF-α [213] and IL-1β [214], to the uninjured sciatic nerve trunk, provokes ectopic antidromic activity in C-fibers. ATP, serotonin, and norepinephrine also induce similar changes in C-fibers [215]. All of these changes result in neurogenic inflammation characterized by increased blood flow, edema, cellular infiltrates, and increased nociception [109] that spreads into the surrounding area due to antidromic activity in Aδ and C-fibers [216]. Therefore, following a nerve resection, there is an acute increase in local C-fiber activity that mimics nerve stimulation, rather than inhibition of neuronal activity. In the absence of an additional injury or insult such as from growing cancer cells, Schwann cells, the essential partners of PNS neurons for function and axonal migration, act to induce regeneration of nerve fibers. When Schwann cell proliferation is interrupted with mitomycin, a large number of incorrectly projecting axons develops [171].

Thus, Schwann cell activity during nerve injury is required to correctly repair the nerve damage and axon projections as well as to reduce cyst formation and secondary damage. Sensory neuromediators might be involved in the proper functioning of Schwann cells because nerve injury induces SP expression in DRG neurons [217], while distal injured and regenerating axons synthesize CGRP locally and independently [218]. For example, CGRP is expressed in over 90% of regenerating branches, irrespective of the expression of their cell bodies. In uninjured sensory neurons, the expression of CGRP is approximately 50% [219]. Furthermore, CGRP, in addition to its vasoactive and immunomodulatory actions, also acts as a Schwann cell mitogen [220] and increases IL-1β and IL-6 expression in Schwann cells [221]. Together, these findings suggest that CGRP and SP may play vital roles in both the Wallerian degeneration and regeneration processes [222,223].

It is also important to note here that capsaicin may affect Schwann cell activity. Capsaicin was recently shown to have immunomodulatory and anti-oxidative effects on activated Schwann cells. Capsaicin reduces antigen presentation, increases resistance to oxidative stress, and enhances the expression of an anti-inflammatory profile [224]. These results are in concordance with previous data showing an anti-inflammatory effect of capsaicin, as discussed above.

Finally, although activation of Schwann cells by cancer cells resembles in many aspects Schwann cell activation during Wallerian degeneration [56,225,226], the regeneration part of this activation and subsequent silencing of Schwann cells is lacking given the presence of the chronic inflammatory microenvironment. Previously, it was shown that metastatic breast carcinoma induces degeneration of C-fibers in areas far from the tumor cells as well as metastatic lesions [156]. Similar changes were reported for in vivo melanoma growth [56]. Therefore, cancer cells are likely to induce degeneration of C-fibers and loss of sensory mediators together with abnormal activation of Schwann cells, leading to dysfunctional/abnormal neurogenesis in the tumor milieu. Injured axons are exposed to Wallerian degeneration following nerve damage. This may lead to the generation of ectopic activity. Wallerian degeneration is controlled by the PNS glial cells’, (i.e., Schwann cells’) activation/dedifferentiation, which is associated with macrophage infiltration and release of pro-inflammatory cytokines and chemokines as well as various growth factors.

## 7. Sensory Nerves and Carcinogenesis: Contradictory Findings

Several studies have reported that inhibition of sensory neuronal activity reduces the development of solid tumors, including prostate [226] and gastric [227] cancers, basal cell carcinoma [228], and melanoma [103,229]. Chemical denervation of neurons by benzalkonium chloride decreases tumorigenesis in a rat model of gastric cancer [230]. Similarly, the progression of pancreatic cancer increases via sensory nerves [231,232]. Inactivation of sensory neurons neonatally by capsaicin treatment increases pancreatic tumor formation in the genetically engineered mouse model that includes a pancreas-targeted KRAS gain-of-function mutation and deletion of p53 [233]. Systemic inactivation of sensory neurons increases lung and heart metastasis in breast carcinoma [12,21,22]. Accordingly, vagotomy increases tumorigenesis in a metastatic pancreatic cancer mouse model. In this model, a muscarinic agonist given systemically decreases this effect of vagotomy [234].

On the other hand, the negative impact of sensory nerve fibers on carcinogenesis and metastasis has also been reported. For example, in the xenograft and genetically engineered mouse models of pancreatic cancer, it was found that denervation of capsaicin-sensitive sensory neurons decreases pancreatic intraepithelial neoplasia progression and increases survival in the early stage of cancer [233]. Moreover, models of cervical carcinoma, head and neck squamous cell carcinoma, and serous ovarian carcinoma have also demonstrated dependence on sensory nerves [235,236,237]. Sensory nerves richly innervate bone and are a component of the bone microenvironment. Malignant cells colonizing bone can promote sensory neoneurogenesis and provoke sensory nerves by creating pathological acidosis in the bones. Denervation of sensory nerves or inhibition of sensory nerve excitation has been shown to reduce cancer progression and improve survival in preclinical models [104].

Three main issues should be considered when exploring the role of sensory neurons in cancer growth and metastasis. First, the models in which nerve inactivation prevents or decreases cancer formation often utilize relatively non-aggressive tumors or focus on the initiation and progression of cancer using a carcinogen or genetically engineered mice. Experimental tumors in these models are open to immune editing, rendering them less aggressive and barely metastatic within the duration of the experiments. Furthermore, these models are likely to deal with less inflammatory tumors than advanced carcinomas. Examples of these models are basal cells [228], gastric [227,230], and prostate carcinoma [226]. Furthermore, in other studies, chemical sympathectomy by intraperitoneal injection of the neurotoxin 6-hydroxydopamine hydrobromide decreased transplantable highly aggressive B16F10 melanoma [229] and BP6-TU2 fibrosarcoma cells [238]. The sympathetic system in most cases opposes the effects of the parasympathetic system, which is mainly formed by the vagus nerve. Hence, inhibition of sympathetic activity is somewhat similar to the activation of the parasympathetic system, which also closely interacts with sensory neurons (cholinergic-peptidergic reciprocal interactions) as explained above and in Kessler (1985) and Ahren et al. (1986) [239,240].

The second issue is the duration of the nerve inactivation, the reactive ectopic neuronal activity, and the effects of these secondary changes on cancer growth and metastasis. For example, several days following sectioning of the L5 spinal nerve, almost all of the cutaneous C-fibers within the L4 spinal nerve displayed a high incidence of spontaneous activity [241,242]. This activity was present within as little as one day following the nerve injury, increased over time, and lasted for several weeks. Consequently, there is excessive local activation of C-fibers after nerve transection lasting for several weeks. This creates a local acute inflammatory response. Injecting cancer cells that are not very aggressive and sensitive to immune editing within these acute inflammatory environments may lead to the elimination of cancer cells. The opposite occurs with aggressive orthotopic syngeneic models, such as breast carcinoma models with 4T1 cells and metastatic derivatives, as well as melanoma models with B16F10 cells. These malignant cells secrete chemokines and cytokines, which are likely to enhance local responses to nerve injury, leading to an immunosuppressive inflammatory microenvironment. Similarly, subdiaphragmatic vagotomy resulted in increased tumor growth and impaired survival in the murine orthotopic pancreatic cancer model; this was associated with increased inflammation [121].

If the duration of nerve injury before the carcinogenic insult is longer than several weeks, then adaptive neuronal transformations are likely to occur, of which some changes may have antitumor effects. It was shown, for instance, that the detrimental effects of vagotomy on acute experimental colitis decreased with longer intervals between surgical vagotomy and stimulation of colitis. This demonstrates that compensatory changes occur after vagotomy, leading to anti-inflammatory processes, (e.g., IL-10 secretion and regulatory T cell induction) [243,244]. When experiments are designed to specifically inactivate C-fibers, secondary acute and subacute inflammatory responses do not occur [245,246,247,248,249], and, thus, these models are likely to reflect the direct effect of nerve inactivation on tumor growth and metastasis. In fact, it was shown that perivagal capsaicin treatment, similar to vagotomy, enhances breast cancer metastasis, demonstrating a protective role of capsaicin-sensitive C-fibers in cancer [56].

The third issue is related to local versus systemic effects of sensory nerve activity. Studies examining the effects of sensory nerve activity on systemic dissemination of the cancer demonstrated that metastasis increases in the absence of systemic sensory nerve activity or vagus nerve activity while the rate of primary tumor growth is not altered [56]. Local effects of sensory activity may vary due to heterogeneity of the sensory nerve fibers in a specific location as well as the diversity of concomitantly present other neuronal/immunological factors within the tumor microenvironment. Specifically, eleven fundamentally distinct types of sensory neurons were discovered and validated by immunohistochemical staining. At the pathological point, it was also shown that pruritus during inflammatory skin diseases might be linked to a distinct itch-generating type [250,251]. Accordingly, seven subtypes of colonic sensory neurons using single-cell RNA-sequencing have been described recently [252]. In addition to this diversity of sensory neurons, diversity in innervation patterns of organs also exists. For example, certain organs, such as the colon and pancreas, are extensively innervated by a wide variety of neurons [253,254], and the final local effects of sensory neuromediators might be shaped by other neuronal factors.

## 8. Conclusions

The interplay between the immune and nervous systems in the tumor microenvironment has been acknowledged in the past, but only more recent studies have started to unravel the molecular players of such complex and dynamic interactions [1]. Some of these studies have reported that excision or chemical ablation of different types of nerves, including sensory fibers, reduces tumor progression in a tissue-specific manner [255]. At the same time, sensory denervation has been reported to accelerate metastasis formation in several preclinical models or to increase the risk of cancer development in clinical studies. These contradictory and challenging results raise important questions about the fundamental subject of tumor innervation, including the origin of tumor-associated neurons, mechanisms of nerve ingrowth (neoneurogenesis) or axonogenesis, their functionality and the process of degeneration, their role in creating the tumor microenvironment, and their contribution to malignant cell survival, motility, and dissemination. Furthermore, the function of intratumoral and/or peritumoral innervation relies on the joint efforts of peripheral neurons and glial cells, whose crosstalk is essential for the integration of their activity in the tumor milieu. Investigation of the signaling mechanisms and delineation of the longitudinal aspects of neuron-to-glia communication at the tumor site are important next steps in the field. Finally, the addition of new players, such as immune-effector and regulatory cells, cancer-associated fibroblasts, tumor-associated platelets, other stromal elements, and tumor-neuron units, should undoubtedly bring new insights to bear on our understanding of tumor invasiveness and help in the formulation of new research objectives and new therapeutic goals.

## Figures and Tables

**Figure 1 cancers-14-02333-f001:**
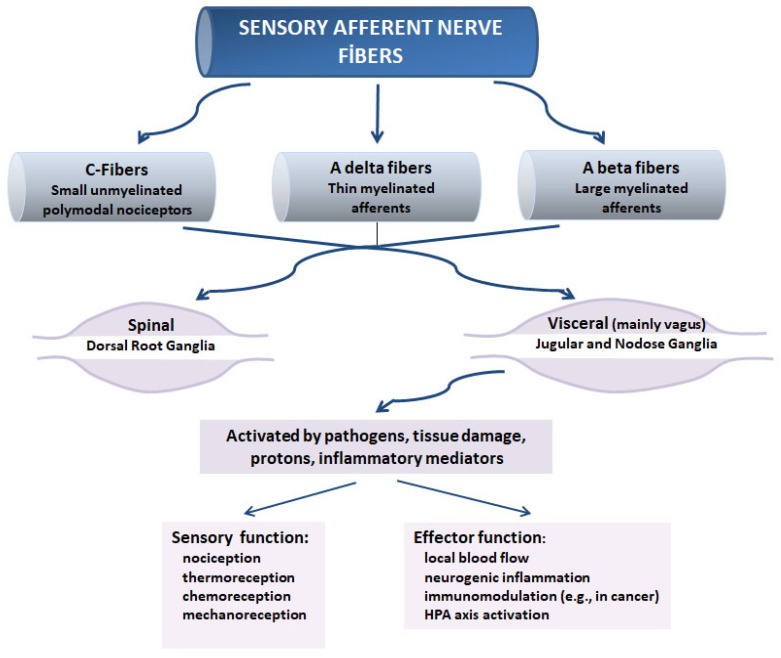
Anatomical and functional classification of sensory nerve fibers.

**Figure 2 cancers-14-02333-f002:**
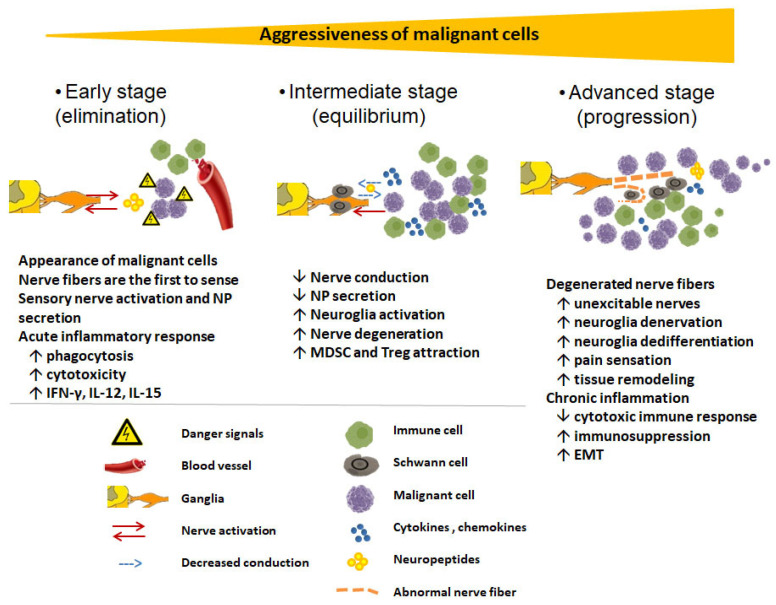
Hypothetical role of sensory nerve fibers in the cancer immune response based on published data. Cancer cells go through immune editing and become more aggressive, while nerve fibers and Schwann cells go through cancer cell editing and become tumorigenic. Free sensory nerve endings of visceral and peripheral tissues are the first to sense the danger signals from pathogens and abnormal cells, inducing an acute inflammatory response as well as the compensatory anti-inflammatory pathway through the hypothalamic–pituitary–adrenal axis in the early stage. In the intermediate stage, partly functioning nerve fibers provide local anti-inflammatory-response cytokines/chemokines released by activated Schwann cells, enhancing cytotoxic immunity. In the advanced stage, complete degeneration of nerve fibers followed by formation of abnormal nerve fibers and tumorigenic Schwann cell activation markedly enhances chronic inflammation, and MDSCs impair cytotoxic immunity and induce metastasis.

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
