# Peer review of "Regulation of Carcinogenesis by Sensory Neurons and Neuromediators"

_cancers, 2022, doi:10.3390/cancers14092333_

Round 1
Reviewer 1 Report
The manuscript entitled: “Regulation of carcinogenesis by sensory neurons and neuromediators”, appears to be carefully done.
This paper suffers from some shortcomings and should be incorporated in a revision of this manuscript:
- The Reference part should be cited more last time published papers. Please revise the Reference.
Author Response
- The Reference part should be cited more last time published papers. Please revise the Reference.
Response: References were updated and also extensive revisions were included as suggested by reviewer 2
Reviewer 2 Report
The manuscript by Erin N et al aims to summarize available literature regarding the influence of sensory neurons and various neuromediators on cancer pathogenesis, with a somewhat specific focus for neuro-immunological interactions.
Although their summary and abstract mainly implicates that the manuscripts discusses sensory neurons in cancer a such, the authors also include glial cells and focus on different aspects that are of importance of the carcinogenic process like inflammation and tumor immunity. Nevertheless, I believe the authors can improve their manuscript by reorganizing their manuscript and rewriting parts to make a really coherent story line: i.e. removing several “non-cancerous-related knowledge” which allows them to combine several paragraphs and specifically focus on “cancer-related knowledge”. I’m convinced that this will enhance readability of the manuscript a lot (suggestion depicted in the major concern section). Also, while some “terms” are really explained in-depth, which is unneeded, some other terms are not thoroughly explained, even though these are used “in a similarity” with the carcinogenic process. Please pay attention to these things (e.g. Wallerian degeneration). Finally, while the authors do take the glial cells into play, these are not mentioned within the introduction, neither are any other of the cell types that have been mentioned in the conclusion, as potential cells to look into in order to fully comprehend carcinogenesis. Consequently, I would like to suggest to write a more general introduction, not only focusing on sensory neurons, but also a bit on their important neuropeptides and on the tumor-microenvironment.
Major concerns:
- While the summary, abstract and introduction really focus on sensory neurons in cancer pathogenesis, the rest of the paper also focusses on other aspects that are of importance for carcinogenesis. Therefore, I would like to suggest to rewrite / reorganize the summary/abstract and especially the introduction, including more of the components / topics that will be discussed. The summary and abstract end with the fact that more research is needed to dive into the neuro-immunology, but the sentence above do not steer towards this message. Also, the authors very comprehensively depict knowledge regarding sensory neurons, but they do not introduce glial cells or immunity in cancer, nor the fact that several mediators can be applied to alter sensory neuronal functioning/physiology. As a consequence, it is – though pleasantly – surprising to have the following topics discussed underneath similar subheading without a proper introduction.
- The authors depict “contradictory findings” can be explained by… but are these really contradictory findings as it is not complete about the same type of neurons having this particular influence on cancer ?
- The authors use the keyword of neo-neurogenesis, but this word is not at all apparent form the summary, neither from the abstract.
- I would suggest to combine subheading 2+3 into one part – because I do not see the added value of this first part alone, especially because here the authors already discuss about inflammation etc., but only in part 3, they state that these neurons respond to pathogens. Therefore, I would like to suggest to start off with the general part about the nervous and immune system in cancer and then come up with the role of sensory neurons and the receptors, that also can be blocked and how this would be beneficial (or not). Also, since it is unclear while different topics besides cancer are touched upon, as they do not specifically add to the pathogenicity of these neurons / neuromediators in the context of carcinogenesis, I would suggest to remove these parts.
- Within section 4, the authors tried to combine a lot of different information into once paragraph, but it doesn’t seem to be a coherent story. I believe some irrelevant information is interspersed within the actual parts really depicting a role in immunity and anti-tumor activity. Therefore, I would like to recommend reorganizing this part, removing all irrelevant information (or at least summarizing in a couple of sentences) so that readers are not devoid from the main message of this paragraph.
- I would like to see that the following subheading would be named (sensory neuron) neuromediators and under these depict SP, CGRP – because now it seems a sudden change from the sensory neurons to a specific neuromediators.. and then immediately add their function within the context of cancer again, in stead of providing big parts of irrelevant information. So please try to summarize the information needed to explain their role in the context of cancer, but remove the parts that are not adding to the story, this will really make sure that the story becomes better understandable for the broad readership.
- Please combine part 9+10 into one section – I believe this is a wonderful opportunity to combine the data depicted in 9 +10 regarding the different models and varying outcomes and immediately debate about the differences that have been observed and to what these can be ascribed too.
- Suggestion to replace part 11 prior to part 9+10 – because this part more belongs to the mechanisms that contribute to the pathogenesis of carcinogenesis and is less focused on the “therapeutic” and blocking strategies and their different outcomes that have been reported.
Minor concerns:
- Please adapt the following words/sentences
a. summary
1)… entire body. They … signals, including those …
2) Various studies have demonstrated that inactivation of …
3).. fibers, as well as the vagus nerve, …
4) Please specify models in line 4
5) … there are also contradictory findings that…
6) These discrepancies are likely caused by …
b. abstract:
1) Please remove the from line 2+3, as well as now and still
2) Breast carcinoma as well as pancreatic
3) Hence, ….
c. Keywords
1) Neuro-immunology
d. Introduction
1) These nerve endings … what is meant by this sentence – where does “these” refer to?
2) Neuropeptide expression in DRGs instead of dorsal root ganglia
3) Nerve injury also represents
4) Please specify effector functions in the final sentence of the introduction
e. Capsaicin-sensitive sensory neurons
1) Reverse sentence one, to show that here you will be talking about altering the function/physiology of sensory neurons
2) Refer back to the A/C fibers that have been described in the introduction for clarity – because these fibers come back again in the following parts
3) First state the sentence about capsaicin at low doses… and following the sentence about activation of the TRPV1 receptor…
4) Again, specify effector functions (providing some examples as such, not with a whole lot of specifications)
5) How come it is obvious that capsaicin only depletes the neuromediators within the nerve endings and not in the CNS, please specify
f. Part 3
1) From which axons are these neuromediators released, sensory ones or ?
g. Part 4-12
1) I have been in the cancer field for about 8 year now and I have never heard anyone specifying cancer as “wounds that do not heal” – could you elaborate and rephrase accordingly
2) Please be consistent in using DRG instead of dorsal root ganglia
3) Always use a “,” after hence
4) Does this neonatal blocking of the sensory nerves also have some other complications?
5) Can neurogenesis be explained in more details and what is meant by the authors – isn’t it all about axonogenesis in these studies, or is really the formation of new nerve (cell bodies).
6) Why could these other components shed more light on the pathogenesis of cancer.. please try to be a bit more specific in this regard (maybe go into neuromediators that signal to these cell types).
Author Response
Major concerns:
- While the summary, abstract and introduction really focus on sensory neurons in cancer pathogenesis, the rest of the paper also focusses on other aspects that are of importance for carcinogenesis. Therefore, I would like to suggest to rewrite / reorganize the summary/abstract and especially the introduction, including more of the components / topics that will be discussed. The summary and abstract end with the fact that more research is needed to dive into the neuro-immunology, but the sentence above do not steer towards this message. Also, the authors very comprehensively depict knowledge regarding sensory neurons, but they do not introduce glial cells or immunity in cancer, nor the fact that several mediators can be applied to alter sensory neuronal functioning/physiology. As a consequence, it is – though pleasantly – surprising to have the following topics discussed underneath similar subheading without a proper introduction.
Response: We included an introduction focusing on neuro-immune interactions before the main text and also a separate introduction for part discussing Schwann cells. Abstract was also modified as requested. All the changes were highlighted in red. New and updated references were added all through the text. We greatly appreciated all the comments of the reviewer, which improved our review markedly. We also reorganized the review; combining sections as requested.
- The authors depict “contradictory findings” can be explained by… but are these really contradictory findings as it is not complete about the same type of neurons having this particular influence on cancer ?
Response: We agree with the reviewer such that local effects of sensory activity may vary due to heterogeneity of the sensory nerve fibers in a specific location as well as diversity of concomitant presence of other neuronal/immunological factors within tumor microenvironment.
This part was further explained in section 7 and highlighted in red.
- The authors use the keyword of neo-neurogenesis, but this word is not at all apparent form the summary, neither from the abstract.
Response: We removed the key word.
- I would suggest to combine subheading 2+3 into one part – because I do not see the added value of this first part alone, especially because here the authors already discuss about inflammation etc., but only in part 3, they state that these neurons respond to pathogens. Therefore, I would like to suggest to start off with the general part about the nervous and immune system in cancer and then come up with the role of sensory neurons and the receptors, that also can be blocked and how this would be beneficial (or not). Also, since it is unclear while different topics besides cancer are touched upon, as they do not specifically add to the pathogenicity of these neurons / neuromediators in the context of carcinogenesis, I would suggest to remove these parts.
Response: We combined the subheading 2+3. Because direct studies regarding the role of sensory neurons in cancer immune response are limited, we want to keep the information regarding to the immune response to pathogens since those studies document their role in cytotoxic immunity.
- Within section 4, the authors tried to combine a lot of different information into once paragraph, but it doesn’t seem to be a coherent story. I believe some irrelevant information is interspersed within the actual parts really depicting a role in immunity and anti-tumor activity. Therefore, I would like to recommend reorganizing this part, removing all irrelevant information (or at least summarizing in a couple of sentences) so that readers are not devoid from the main message of this paragraph.
Response: This part was reorganized and shorten (from 1052 words to 687 words)
- I would like to see that the following subheading would be named (sensory neuron) neuromediators and under these depict SP, CGRP – because now it seems a sudden change from the sensory neurons to a specific neuromediators.. and then immediately add their function within the context of cancer again, in stead of providing big parts of irrelevant information. So please try to summarize the information needed to explain their role in the context of cancer, but remove the parts that are not adding to the story, this will really make sure that the story becomes better understandable for the broad readership.
Response: We reorganized this part and highlighted in red. This part was shorten from 1581 to 1430 words to remove irrelevant information.
Please combine part 9+10 into one section – I believe this is a wonderful opportunity to combine the data depicted in 9 +10 regarding the different models and varying outcomes and immediately debate about the differences that have been observed and to what these can be ascribed too.
Response: We combined part 9+10 (now it is last part (part 7) before conclusions, reorganized and extended the discussion upon reviewer’s comments (all highlighted).
- Suggestion to replace part 11 prior to part 9+10 – because this part more belongs to the mechanisms that contribute to the pathogenesis of carcinogenesis and is less focused on the “therapeutic” and blocking strategies and their different outcomes that have been reported.
Response: We have rearranged all the parts according to the comments and also included and introduction to Schwann cell section removing blocking strategies.
Minor concerns:
- Please adapt the following words/sentences
- summary
1)… entire body. They … signals, including those …
2) Various studies have demonstrated that inactivation of …
3).. fibers, as well as the vagus nerve, …
4) Please specify models in line 4
5) … there are also contradictory findings that…
6) These discrepancies are likely caused by …
Response: All corrected and highlighted in red
- abstract:
1) Please remove the from line 2+3, as well as now and still
2) Breast carcinoma as well as pancreatic
3) Hence, ….
Response: Corrected and highlighted in red
- Keywords
1) Neuro-immunology
Response: Corrected
- Introduction
1) These nerve endings … what is meant by this sentence – where does “these” refer to?
Response: These refer to sensory- corrected
2) Neuropeptide expression in DRGs instead of dorsal root ganglia
3) Nerve injury also represents
4) Please specify effector functions in the final sentence of the introduction-
Response: All corrected and effector functions are specified
- Capsaicin-sensitive sensory neurons
1) Reverse sentence one, to show that here you will be talking about altering the function/physiology of sensory neurons.
Response: We rearranged the first sentence
2) Refer back to the A/C fibers that have been described in the introduction for clarity – because these fibers come back again in the following parts.
Response: We included required change and highlighted in red.
3) First state the sentence about capsaicin at low doses… and following the sentence about activation of the TRPV1 receptor…
Response: We included required change and highlighted in red.
4) Again, specify effector functions (providing some examples as such, not with a whole lot of specifications).
Response: We included required change and highlighted in red.
5) How come it is obvious that capsaicin only depletes the neuromediators within the nerve endings and not in the CNS, please specify
Response: This was shown before by Helke et al. 1981 PMID: 6169396
- Part 3
1) From which axons are these neuromediators released, sensory ones or ?
Sensory, the following previous sentence that was present in the text explains it:
Response: Specifically, neuropeptides such as calcitonin gene‐related peptide (CGRP) and substance P (SP) are synthesized in unmyelinated sensory neurons and released from their peripheral terminals upon activation by local factors.
- Part 4-12
1) I have been in the cancer field for about 8 year now and I have never heard anyone specifying cancer as “wounds that do not heal” – could you elaborate and rephrase accordingly
Response: As far as we know this phrase was first described in 1986 by H.F Dvorak and it describes the pathology from a different perspective. :
H.F Dvorak. Tumors: wounds that do not heal: similarities between tumor stroma generation and wound healing. N Engl J Med 1986; 315: pp. 1650-1659.
Dvorak, H.F. Tumors: wounds that do not heal-redux. Cancer Immunol. Res. 2015, 3, 1-11, doi:10.1158/2326-6066.CIR-14-0209.
Byun JS, et al, Wounds that will not heal: pervasive cellular reprogramming in cancer. Am J Pathol. 2013.
2) Please be consistent in using DRG instead of dorsal root ganglia
Response: Corrected
3) Always use a “,” after hence
Response: Corrected
4) Does this neonatal blocking of the sensory nerves also have some other complications?
Response: Neonatal blocking of the sensory nerves have some complications such as changes in behavioral development and consequently formation of skin ulcer (PMID: 9511172, PMID: 8090813). These complications however do not directly relate to carcinogenesis and immune response.
5) Can neurogenesis be explained in more details and what is meant by the authors – isn’t it all about axonogenesis in these studies, or is really the formation of new nerve (cell bodies).
Response: Both axonogenesis and neurogenesis are reported (PMID: 19047084, PMID: 31092925) but we agree it is mostly axonogenesis
Round 2
Reviewer 2 Report
Revised review – Cancers manuscript – Erin N eta al: Regulation of carcinogenesis by sensory neurons and neuromediators
I’m very pleased to see that the authors have incorporated all suggestions that I’ve made and answered all the questions that I had.
Now their manuscript indeed focusses not only on sensory neurons, but also on glial cells and different aspects involved in the carcinogenic process like inflammation and tumor immunity. Given that the introduction now addresses already all different components that will discussed, even as the combination of several of the paragraphs into 1, removing al non-relevant information, I truly believe that the manuscript now reads fluently and has become a lot more clear for all readers, even those coming from different research fields.